# Molecular Characterization of Non-Neurogenic and Neurogenic Lower Urinary Tract Dysfunction (LUTD) in SCI-Induced and Partial Bladder Outlet Obstruction Mouse Models

**DOI:** 10.3390/ijms24032451

**Published:** 2023-01-26

**Authors:** Michelle von Siebenthal, Akshay Akshay, Mustafa Besic, Marc P. Schneider, Ali Hashemi Gheinani, Fiona C. Burkhard, Katia Monastyrskaya

**Affiliations:** 1Functional Urology Research Laboratory, Department for BioMedical Research DBMR, University of Bern, 3008 Bern, Switzerland; 2Graduate School for Cellular and Biomedical Sciences, University of Bern, 3012 Bern, Switzerland; 3Department of Urology, Inselspital University Hospital, University of Bern, 3010 Bern, Switzerland

**Keywords:** bladder remodeling, urodynamic investigation (UDI), external urethral sphincter (EUS), electromyography (EMG), transcriptome, dyssynergia, spinal cord injury (SCI), partial bladder outlet obstruction (pBOO)

## Abstract

We examined bladder function following spinal cord injury (SCI) by repeated urodynamic investigation (UDI), including external urethral sphincter (EUS) electromyography (EMG) in awake restrained mice and correlated micturition parameters to gene expression and morphological changes in the bladder. A partial bladder outlet obstruction (pBOO) model was used for comparison to elucidate both the common and specific features of obstructive and neurogenic lower urinary tract dysfunction (LUTD). Thirty female C57Bl/6J mice in each group received an implanted bladder catheter with additional electrodes placed next to the EUS in the SCI group. UDI assessments were performed weekly for 7 weeks (pBOO group) or 8 weeks (SCI group), after which bladders were harvested for histological and transcriptome analysis. SCI mice developed detrusor sphincter dyssynergia (DSD) one week after injury with high-pressure oscillations and a significantly increased maximal bladder pressure P_max_ and were unable to void spontaneously during the whole observation period. They showed an increased bladder-to-bodyweight ratio, bladder fibrosis, and transcriptome changes indicative of extracellular matrix remodeling and alterations of neuronal signaling and muscle contraction. In contrast, pBOO led to a significantly increased P_max_ after one week, which normalized at later time points. Increased bladder-to-bodyweight ratio and pronounced gene expression changes involving immune and inflammatory pathways were observed 7 weeks after pBOO. Comparative transcriptome analysis of SCI and pBOO bladders revealed the activation of Wnt and TGF-beta signaling in both the neurogenic and obstructive LUTD and highlighted FGF2 as a major upregulated transcription factor during organ remodeling. We conclude that SCI-induced DSD in mice leads to profound changes in neuronal signaling and muscle contractility, leading to bladder fibrosis. In a similar time frame, significant bladder remodeling following pBOO allowed for functional compensation, preserving normal micturition parameters.

## 1. Introduction

Patients with lower urinary tract dysfunction (LUTD) experience a variety of lower urinary tract symptoms, including urinary frequency, urgency, nocturia, hesitancy, urinary incontinence, slow or intermittent stream, incomplete bladder emptying, and terminal dribble [1]. Frequent urinary tract infections, sleep cycle disturbances, and debilitating incontinence affect the physical and emotional well-being of patients and greatly impair their quality of life [2]. The socioeconomic burden caused by LUTD is significant since the healthcare costs in the USA were more than 2.5 times higher for patients with an overactive bladder than for similar patients without overactivity [3]. 

LUTD has multiple causes, including aging, diabetes, various forms of bladder outlet obstruction (BOO), and neurological diseases. One common cause of non-neurogenic LUTD is benign prostatic hyperplasia, leading to benign prostatic obstruction (BPO). Neurological diseases, including spinal cord injury (SCI), Parkinson’s disease, and multiple sclerosis, may lead to neurogenic LUTD (NLUTD). Depending on the level of the central nervous system lesion, detrusor sphincter dyssynergia (DSD), a discoordination between the bladder smooth muscle (detrusor) and the external urethral sphincter (EUS) muscles occurs, resulting in a functional obstruction when both contract simultaneously when attempting to void. Urodynamic investigation (UDI) is the only method to objectively assess bladder function. Increased bladder pressure, post-void residual, and detrusor over- or underactivity may be observed in BLUTD and NLUTD. Consequently, the bladder tissue remodels and bladder inflammation progresses to hypertrophy and to fibrosis, which ultimately leads to bladder decompensation or failure if left untreated [4]. 

Comprehensive analysis of differentially expressed mRNAs and microRNAs (miRNAs) in bladder biopsies from BOO patients using next-generation sequencing (NGS) revealed the involvement of cytokine and immune response, hypertrophic phosphatidylinositol 3-kinase/AKT, nitric oxide and transforming growth factor β (TGF-beta) signaling pathways [5]. Furthermore, dysregulation of transcription regulators and effectors associated with inflammation (tumor necrosis factor α (TNF-α), nuclear factor ‘kappa-light-chain-enhancer’ of activated B-cells (NF-kB) and AP-1), hypoxia (hypoxia-inducible factor-1α (HIF1α)), proliferation, oxidative stress, and contractility were identified [5,6]. Experimental partial BOO (pBOO) in mice is frequently used to model the longitudinal changes in bladder morphology and function. pBOO is accompanied by an upregulated mRNA expression of pro-fibrotic genes, inflammation markers, hypoxia-inducible factors, inducible nitric oxide synthetase (iNOS), epithelial–mesenchymal transition-inducing transcription factors, and muscle contractility proteins (α-SMA and SM22) and regulators (RhoA and ROCK-β) [7,8,9,10,11,12,13] and downregulation of some muscle contractility regulators (caveolins) [14]. A recently published study in rats [15] combined RNA sequencing and TMT-labeling proteomic analyses to study molecular changes after 4 and 5 weeks of pBOO. Partial BOO in rats leads to bladder hypertrophy accompanied by activation of focal adhesion at the cell-extracellular matrix contact points, regulation of actin cytoskeleton, smooth muscle contraction, gap junctions, and cell cycle pathways at week 4, and increasing involvement of immune response pathways at week 5 of obstruction [15]. 

Information concerning the changes in the bladder transcriptome following SCI is very limited. To date, only one study examined mRNA expression in the detrusor of patients with neurogenic detrusor overactivity, which included patients with SCI, spina bifida, and cerebral palsy. RT-qPCR experiments revealed a decreased large conductance Ca^2+^-activated K^+^ (BK) channel mRNA level [16]. Further studies examined bladder mRNA expression in a mouse model of SCI, showing an altered expression of smooth muscle myosin heavy chain [17] and a decreased expression of Pdgfra and Kcnn3 [18]. Pdgfrα^+^ interstitial cells restrain the excitability of smooth muscle at low levels; thus, the authors concluded that the downregulation of Pdgfra (at mRNA and protein levels) is responsible for the development of detrusor overactivity following SCI [18]. Using NGS, differences in miRNA expression in SCI-induced rat neurogenic bladders were reported, but the authors did not examine the rest of bladder transcriptome [19]. 

Here, we used the previously established mouse model, which allows unanesthetized, repeated UDIs, including EUS-electromyography (EMG) recordings [20], to study changes in bladder function in a mouse model of SCI created by complete transection of the spinal cord at thoracic level 8 (T8). We compared the data with a pBOO mouse model obstructed for a comparable period of time (8 vs. 7 weeks). Bladder function was examined by weekly UDIs, including EUS-EMG in SCI mice, and was correlated with gene expression.

## 2. Results

### 2.1. Study Design and Animal Grouping

Experiments were performed using C57Bl/6J mice (Figure 1). In the SCI study (Figure 1A) (*n* = 30), a catheter was implanted in the bladder, two electrodes next to the EUS and one electrode into the abdominal muscle. Three mice did not recover from the implantation surgery. One week later, mice were randomly allocated to the study groups sham (*n* = 8) and SCI (*n* = 19), and a baseline UDI was recorded. Then, mice according to their group allocation underwent either sham operation or complete transection of the spinal cord at thoracic level 8 (T8). After surgery, UDI was performed weekly for 8 weeks. Thereafter, mice were sacrificed and bladders were harvested for histology and total RNA isolation for NGS and transcriptome analysis. 

In the pBOO study (*n* = 30), a catheter was implanted in the bladder (Figure 1B). Three mice did not recover from the implantation surgery. After one week, mice were randomly allocated to one of the four study groups: “sham 1 week” (*n* = 5), “pBOO 1 week” (*n* = 8), “sham 7 weeks” (*n* = 5), and “pBOO 7 weeks” (*n* = 9). A baseline UDI was recorded in all animals. Mice belonging to a pBOO group (“pBOO 1 week” and “pBOO 7 weeks”) underwent pBOO surgery, and mice belonging to a sham group (“sham 1 week” and “sham 7 weeks”) underwent sham surgery. A UDI was performed 1 week after surgery in the “pBOO 1 week” and “sham 1 week” groups. Thereafter, the mice were sacrificed and the bladders harvested for histology and transcriptomic analysis (Figure 1B, 1 week mice). In the “sham 7 weeks” and “pBOO 7 weeks” groups, UDIs were performed once a week for 7 weeks (Figure 1B, 7 weeks mice). The mice were then sacrificed and the bladders harvested for histology and transcriptome analysis. 

### 2.2. SCI in Mice Causes Severe LUTD, Leading to Bladder Fibrosis

Following baseline UDI and sham or SCI surgery, UDIs were performed weekly over 8 weeks. Representative UDI traces are shown in Figure 2 at baseline, week 2, and week 8. Sham animals (*n* = 8) showed normal micturition cycles with distinct filling and voiding phases throughout the entire experiment (Figure 2A). However, the timing of micturition and measuring voided volumes was not always reliable in our UDI model, because it was performed by recording the weight of urine fallen on the scale underneath the mouse. In reality, voids were frequently trapped in the animal’s fur or on the restrainer, and could be visualized but not recorded (V_void_ in baseline, week 2 and week 8 in Figure 2A). In SCI animals (*n* = 12), micturition cycles were not observed starting 1 week (*n* = 9) or 3 weeks (*n* = 3) after injury (Figure 2B). Interestingly, three SCI mice (25%) showed transient micturition cycles at week 2, which disappeared at later time points. During UDI in the SCI mice, intravesical pressure (P_ves)_ increased during bladder filling, then oscillated at a high level, and numerous contractions were observed. However, these contractions failed to efficiently empty the bladder and restore bladder pressure to the baseline levels. In SCI animals, maximal detrusor pressure (P_max)_ was significantly increased at week 1–8 when compared to sham (Table 1). 

UDIs and EUS-EMG measurements were recorded simultaneously. EUS activity during voiding was compared to EUS activity before and after voiding. To circumvent the problem of timing the voids, we resorted to the strategy adopted in our earlier study [20] (see Materials and Methods below): the time from threshold detrusor pressure (P_thresh_) to P_max_ was divided into equal fifths, and the middle (third) fifth was considered the section when voiding occurs. The EUS activity during each section (before, during, and after voiding) is shown as a percentage of the total activity during all three sections (Appendix A). Before surgery (baseline), 5 out of 7 animals (71%) in the sham group and 11 out of 17 animals in the SCI group (65%) showed a v-shaped EUS-EMG activity, demonstrating reduced EMG activity during voiding compared to before and after voiding (Appendix A, baseline). At the later time points (shown for week 1 and week 3), the v-shaped EUS-EMG activity pattern was sustained in 50–66% of sham mice. In SCI animals, when no micturition cycles were observed, the EUS-EMG activity was analyzed during the P_ves_ oscillations. One and three weeks post SCI (Appendix A), only 2 out of 10 (20%) showed a mean v-shaped activity. Overall, the percentage of contractions with a v-shaped EUS-EMG activity in SCI animals ranged from 14% at week 1 to 37% at week 4 (Appendix A). Concomitant with developing neurogenic LUTD, the bladder-to-bodyweight ratio was significantly increased following SCI (*p* = 0.0116, Figure 2C), and the proportion of collagen in the bladder wall was significantly higher than in the sham group (*p* = 0.017, Figure 2C). 

### 2.3. Transcriptome Analysis of SCI Mouse Bladders Reveals Dysregulation of Neuronal and Contractile Processes Accompanied by Fibrosis

NGS was used to analyze the bladder transcriptome 8 weeks after sham or SCI surgery. Out of 21751 genes with nonzero total read counts, 174 genes (0.8%) were significantly upregulated, and 151 genes (0.69%) were significantly downregulated (adjusted *p*-value < 0.1) (Appendix A). The volcano plot in Figure 3A shows the significantly regulated genes in red. The principal component analysis (PCA) of the top 500 regulated genes demonstrated suitable clustering of the SCI samples (Figure 3B). We performed a gene ontology over-representation analysis (GO-ORA) to gain insight into the biological processes underlying pBOO. A semantic similarity matrix depending on the information content of the closest common ancestor term was created for the resulting list of GO terms per comparison. Regulated biological processes are summarized in the treemap of the gene ontology (GO)-term clusters (Figure 3C), showing that muscle cell processes were highly represented in the SCI dataset, based on the abundance of muscle-connected GO terms (muscle system processes, striated muscle cell differentiation, contractile actin filament bundle assembly, actomyosin structure organization, response to muscle stretch, and skeletal muscle contraction). Furthermore, proliferative (regulation of cellular response to growth factor stimulus, regulation of cell growth, and epithelial cell proliferation), extracellular matrix (connective tissue development, extracellular matrix organization, and collagen metabolic process), calcium (regulation of cytosolic calcium ion concentration, positive regulation of calcineurin-mediated signaling, and response to calcium ion), and angiogenesis-related processes (positive regulation of angiogenesis) were regulated following SCI. Noteworthy is that immune- and inflammation-related processes were scarce: positive regulation of B cell and leukocyte activation and complement activation were the only two abundant functional groups identified. In line with these findings, the Reactome pathway analysis revealed significant regulation of extracellular matrix (ECM) remodeling and fibrosis, neuronal, and contractility pathways (Figure 3C). The word cloud of regulated genes (Figure 3D) showed an upregulation of growth factors (FGF9, IGF1), Wnt signaling- (SFRP1, SFRP2), fibrosis- (CCN3, CCN4, Thbs1), and cell migration-related (SLIT2) genes, and a downregulation of muscle- (AKAP6, CASQ2), calcium- (RYR2), proliferation- (SRF), and immune-related (RGS2) genes. 

### 2.4. Changes in Bladder Function and Morphology after pBOO and SCI Are Different 

UDIs were recorded at baseline (before sham or pBOO surgery) and once a week for 7 weeks after surgery. Representative traces of P_ves_ and voided volume (V_void_) during 3 micturition cycles are shown at baseline and weeks 1 and 7 (Figure 4) for the sham-operated (Figure 4A), pBOO week 1 group (Figure 4B) and pBOO week 7 group (Figure 4C). Unlike SCI, pBOO did not lead to a dramatic deterioration of bladder function. In pBOO, animals all micturition cycles consisted of a filling phase, during which P_ves_ increased at a relatively low rate, and a voiding phase, during which the detrusor contracted (sharp increase in P_ves_ followed by return to basal pressure). Not all observed voids could be recorded because they were frequently trapped in the animal’s fur or on the restrainer. P_max_ was significantly increased at week 1 in both pBOO groups compared to their respective sham group (Figure 4B) (1 week: *p* = 0.006; 7 weeks: *p* = 0.032) and compared to baseline recording (“pBOO 1 week”: *p* < 0.0001; “pBOO 7 weeks”: *p* = 0.001) (Table 2). 

In the 7 weeks pBOO group, this difference gradually disappeared, and P_max_ was similar to the respective sham (Figure 4C). Fibrotic organ remodeling following pBOO was assessed by Masson’s trichrome stain (Figure 4D). Although the ratio of collagen tissue to whole tissue was unchanged between groups (not shown), there was a significant increase in the bladder-to-bodyweight ratio after 7 weeks of pBOO (Figure 4E). This indicates that the bladder underwent morphological changes despite the fact that bladder function returned to normal 7 weeks after pBOO 

### 2.5. Transcriptome of the Mouse Bladders Progressively Deteriorates from One to 7 Weeks of pBOO 

The bladder transcriptome of pBOO mice was analyzed by NGS (Appendix A). The differentially expressed genes (DEGs) are presented in volcano plots (Figure 5A), showing the up- and downregulated genes defined in pair-wise comparisons between the week 1 and week 7 groups, indicating an increase in the number of DEGs vs. respective shams as obstruction progressed. PCA analysis of the top 500 significant high-variance genes showed distinct clusters of samples, with clear separation of the “pBOO 7 weeks” samples from the rest of the cohort (Figure 5B). GO-ORA showed that one week after pBOO (sham 1 week vs. pBOO 1 week), only a few biological processes appeared in the treemap of the GO-term clusters, including proliferation (cell proliferation involved in heart morphogenesis) and immune response pathways (immunoglobulin production) (Figure 5C). Seven weeks after pBOO (sham 7 weeks vs. pBOO 7 weeks), gene expression changes were massive (Figure 5D). 

The treemap of the GO-term clusters revealed regulation of immune response pathways, inflammation pathways, proliferative pathways, ECM remodeling, muscle-specific pathways, and oxidative stress. Reactome pathways built used 7-week pBOO DEGs revealed regulation of immune response, ECM remodeling, and nuclear signaling (Figure 5E). These changes indicate that in order to compensate for the obstruction, the bladder underwent significant remodeling, including activation of proliferative and contractile processes. Activation of immune responses, detected by NGS, was confirmed by RT-qPCR (Appendix A). A total of 7 weeks after pBOO, Ccl2 and macrophage markers (Adgre1, CD68 and Ccr2) were significantly upregulated (*p* = 0.029, *p* = 0.0028, *p* = 0.0018, and *p* = 0.0182). Conversely, Ccl2 was significantly downregulated 1 week after pBOO (*p* = 0.0014). 

### 2.6. Comparative Transcriptome Analysis after SCI and pBOO Reveals Differences between Neurogenic and Obstructive LUTD 

Despite the compensated bladder function 7 weeks after pBOO compared to the deterioration of micturition and development of neurogenic LUTD 8 weeks after SCI, there were more DEGs with a significant adjusted p-value in pBOO (574) than in SCI (326) bladders (Figure 6A). PCA revealed a clear separation of pBOO and SCI samples (Figure 6B), indicative of diverging molecular composition. 

We used an enrichment map to organize enriched BP terms into a network with edges connecting overlapping gene sets (Figure 6C). As a result, mutually overlapping gene sets tend to cluster together, facilitating the identification of functional modules. In this plot, a circle represents a BP term, and the color indicates whether it is enriched for BOO (black), SCI (blue), or both (yellow). We detected a considerable overlap in shared BPs when comparing BP terms from the two types of LUTD, in particular, the processes of ECM reorganization, ERK signaling, calcium signaling, and TGF-beta-SMAD signaling (Figure 6C). Processes of muscle hypertrophy, actomyosin fibers, and neuron projection, as well as MAPK/Wnt signaling, were specific for SCI-induced NLUTD. After 7 weeks, pBOO was characterized by the activation of immune and inflammatory processes. 

### 2.7. Common and Unique DEGs Define the Molecular Differences between SCI and pBOO Bladders

We compared the DEGs of SCI and pBOO and delineated 59 genes shared between both types of LUTD (Appendix A). Of these, 46 were similarly and 13 oppositely regulated in SCI compared to pBOO. Pathway analysis (Reactome and KEGG) and GO-ORA cellular compartments are shown in Appendix A. Common genes control the pathways of TGF-beta and Wnt signaling involved in collagen biosynthesis, ECM remodeling, and FGFR1 signaling. Cellular components for the common genes are related to ECM, whereas oppositely regulated genes refer to immunoglobulin complex and vesicular structures (Appendix A). 

In order to quantify and visualize the differences between the two types of bladder remodeling (neurogenic vs. non-neurogenic obstruction), we carried out gene enrichment analysis, and then compared the abundance of enriched BP terms with specific keywords across the gene sets. In the radar graph, color represents the gene set, and dot/arm represents the number of enriched BP terms with the given keyword (Figure 7). pBOO unique DEGs are related to BPs involving B and T cells, leukocytes, immune function, proliferation, and differentiation (Figure 7A), reflected in the word cloud of most frequently appearing genes. In contrast, among SCI-specific BPs muscle, contraction, calcium, ion transport, growth, differentiation, and neuron synapse were specifically represented (Figure 7B). The 46 common and similarly regulated DEGs related to muscle, proliferation, growth, and differentiation (Figure 7C). 

Interestingly, SFRP1 and SFRP2 (secreted frizzled-related proteins 1 and 2) are involved in the shared BPs (word cloud, Figure 7C). These proteins are soluble modulators of Wnt signaling and regulate cell growth and differentiation. Fibroblast growth factor 2 (FGF2) is another important regulatory molecule prominently featured among shared DEGs. 

## 3. Discussion

SCI in rodents is a widely used model of NLUTD, which is characterized by a functional BOO caused by DSD. Here, we examined the effects of SCI on bladder function in mice using our previously reported repeated UDI-EUS-EMG recordings in awake restrained mice [20]. In order to monitor detrusor and sphincter activity, we performed UDI-EUS-EMG in mice before and once a week after complete spinal cord transection at the T8 level for 8 weeks. To our knowledge, this is the first study employing repeated EUS-EMG recordings in the same SCI mouse, allowing us to monitor longitudinal changes in bladder function. Following SCI, bladder function was dramatically altered: mice developed urinary retention, and their bladders had to be manually expressed twice a day. During UDI, the loss of bladder function was mirrored by the absence of micturition cycles with distinct voiding phases. The predominant bladder phenotype after SCI was characterized by the rising bladder pressure during filling, which did not result in spontaneous voiding. P_max_ in SCI was significantly higher than P_max_ in sham controls, and the pressure oscillated around that level, with periodic detrusor contractions, not leading to bladder emptying. The uncoordinated EUS-detrusor activity was obvious in the EUS-EMG activity recordings, which did not show the characteristic v-shaped activity (low activity during voiding, high activity before and after voiding). Moreover, EUS-EMG activity tended to increase while P_ves_ increased during contraction, indicative of DSD 

In line with our results, the increasing P_ves_ at the beginning of the recordings followed by high-pressure oscillations and uncoordinated EUS-EMG activity was reported in two previous studies in SCI mice 5 or 18 weeks after complete spinal cord transection at the T8 or spinal contusion at the T10 level [21,22]. In these studies, UDI were performed immediately after catheter and electrode implantation in awake conditions or under terminal urethane anesthesia [21,22]. Kadekawa et al. performed UDI including EUS-EMG in mice 4 weeks after SCI and immediately after catheter and electrode implantation. Similar to our study, they observed no micturition cycles but inefficient, intermittent voiding, which occurred during low-tonic EUS-EMG activity. However, they did not perform any quantification of EUS-EMG activity [23].

Eight weeks after sham/SCI surgery, the bladders were harvested, histologically assessed, and the transcriptome analyzed by NGS. Bodyweight at the end of the experiment was significantly lower in SCI mice due to the muscle loss induced by hind limb paralysis. In contrast, the bladder-to-bodyweight ratio was significantly increased, indicative of bladder remodeling. Indeed, Masson’s trichrome stain revealed a significantly increased collagen deposition in the bladder wall. These findings were supported by the transcriptome analysis: extracellular matrix remodeling, including collagen metabolic processes and upregulation of fibrosis-related genes (CCN3, CCN4), were detected in the bladder. Muscle contractility and neuronal system components were identified as the other main altered BPs. We also detected “response to muscle stretch” among affected processes, which might reflect the urinary retention detected by UDI. This is a disadvantage of rodent SCI models because urinary retention in humans is prevented by catheterization [24]. 

In contrast to neurogenic bladder remodeling caused by SCI-induced DSD, non-neurogenic BOO is often observed in elderly male humans with benign prostatic hypertrophy. The underlying mechanisms of non-neurogenic vs. neurogenic bladder dysfunction appear to be different, resulting in distinct functional and morphological bladder phenotypes. Non-neurogenic BOO is frequently modeled in rodents with pBOO, induced by placing a ligature around the urethra. Variability in the degree of obstruction and high animal mortality are the main disadvantages of this model, in addition to interfering with bladder innervation while dissecting around the proximal urethra and bladder neck [25]. This is reflected by the discrepancy of findings: while many studies show an increased P_max_ after 5 [26,27] and 6 [28] weeks of pBOO, others do not report such changes [29]. Similarly, non-voiding contractions frequently observed after pBOO might be attributed to the artifacts caused by movement. Here we induced pBOO for 1 or 7 weeks in mice and performed longitudinal monitoring of bladder function by UDI. Weekly UDI in the pBOO model up to 7 weeks post-obstruction revealed an initial increase in bladder pressure during voiding at week 1, necessary to empty the bladder against the apparent outlet resistance. In line with these observations, a rapid dysregulation of miRNA and mRNA expression at 10-14 days post pBOO followed by the return to control levels was previously reported in rats [30] and mice [31] and might be an indication of the rapid reaction to outlet obstruction. However, in the following weeks, mice bladders adapted to the obstruction and were able to empty at normal pressure. We hypothesize that the return to normal pressure despite pBOO was possible due to a decrease in urine flow and prolongation of the voiding duration. Unfortunately, our UDI system did not allow us to reliably time the micturition and calculate flow parameters; thus, we could not evaluate whether maintaining physiological P_ves_ led to a reduced flow. 

An increased bladder-to-bodyweight ratio following pBOO in mice was previously reported [26,27,29] and confirmed here 7 weeks after pBOO. The histologic assessment of collagen content by Masson’s trichrome stain revealed no difference between pBOO and sham groups. The increased bladder weight in pBOO might thus have arisen due to urothelial and sub-urothelial thickening caused by hypertrophy or hyperplasia rather than by collagen deposition. Indeed, bladder gene expression analysis showed no dysregulation of genes involved in extracellular matrix remodeling (i.e., collagen deposition) 1 week after pBOO, although the processes involving muscle cell proliferation were highly upregulated at this time point. As obstruction progressed, 7 weeks after pBOO, the upregulation of proliferative pathways persisted, and upregulated genes associated with cell growth (IGF1 and PTPRC) were detected. Additionally, transcriptome analysis revealed ECM reorganization and connective tissue development processes at this time point. Thus, we conclude that fibrotic bladder remodeling might have just started to develop 7 weeks after pBOO and could be detected at the mRNA level before becoming apparent at the structural level (tissue morphology). Overall, alterations in gene expression 1 week after pBOO were small since, apart from proliferation processes, only a few immunoglobulin- (immune response) and oxygen-related genes that might be indicative of a hypoxic condition were upregulated. On the contrary, gene expression was drastically altered 7 weeks after pBOO. Numerous biological processes involved in immune response and inflammation pathways, such as cytokine-mediated signaling, phagocytosis, and interleukin-6 production, were regulated at this time point. Moreover, genes involved in the immune response (CXCL13, SYK, CCR7, CCL5, CD74, and TLR2) and inflammation (CCR2, CX3CL1, and TREM2) were upregulated. RT-qPCR confirmed the upregulation of CCR2 and other markers of macrophage infiltration (ADGRE1, CD68) and inflammation (CCL2) 7 weeks after pBOO. 

Having established and characterized the two types of LUTD in mouse models, we carried out a comparative analysis of the transcriptomes of SCI and pBOO to elucidate the biological processes and function, common and specific for neurogenic and non-neurogenic bladder remodeling. We could not extend pBOO beyond week 7 post-obstruction due to the institutional restrictions caused by the COVID-19 pandemic. The one-week difference in the length of observations between SCI and pBOO is a limitation of this study. Both pBOO- and SCI-induced LUTD led to increased P_max_. While P_max_ was only transiently elevated 1 week after pBOO and normalized thereafter, P_max_ was increased 1 week after SCI and persisted at a high level. PCA revealed a clear separation of samples based on their origin (SCI or pBOO, 8/7 weeks post-surgery), indicating the profound differences in the bladder transcriptomes. Accordingly, muscle-related processes (contractility, hypertrophy, actomyosin fiber organization) were highly regulated in the SCI bladder transcriptome while playing a lesser role after pBOO. We also only obtained molecular evidence of alterations in neuronal projection and synaptic organization in NLUTD, confirming the role of changed innervation in organ dysfunction. The involvement of calcium and other ion transport processes in NLUTD further confirms the increased neuronal and muscle activity. In contrast, the vast majority of DEGs in pBOO were related to immune, inflammatory, and cytokine signaling processes. 

We delineated 59 common genes shared in the SCI and pBOO bladders, with 46 DEGs similarly regulated in both types of LUTD. These 46 genes are involved in the biological processes of muscle contractility, growth, proliferation, and differentiation. Pathways important for collagen formation and ECM remodeling were present in both groups, revealing a common trend in fibrosis. We showed that upregulated FGF2 was involved in both SCI- and pBOO-induced bladder remodeling. Our results confirm previous observations made in rat pBOO, where the elevated bladder and serum FGF2 were detected and implicated in the development of hypertrophy [15]. Fibroblast growth factors (FGFs) induce a contractile-to-proliferative phenotype switch in the smooth muscle and control smooth muscle cell growth and proliferation [32]. FGF-TGF-beta signaling antagonism is the primary regulator of the SMC phenotype switch [33]. In vascular SMC, FGF signaling promotes the contractile to synthetic phenotypic switching and enhances inflammatory response [34]. In addition, activation of FGFR in vascular smooth muscle promoted fibroblast migration, contributing to pathologic vascular remodeling [35]. Similarly, it has been shown to play a major role in pulmonary [36] and hepatic [37] fibrosis. In the bladder, FGFR activation might be indicative of the switch from hypertrophy to fibrotic changes as the result of anatomic or functional outlet obstruction.

Wnt and TGF-beta signaling were the two shared activated pathways in neurogenic and non-neurogenic LUTD. Our previous work demonstrated the importance of Wnt for the maintenance of smooth muscle contractility [38]. Wnt signaling plays an important role in cell proliferation, differentiation, and migration during embryonic development. In adulthood, Wnt signaling contributes to tissue regeneration, and its altered activation has been reported in various pathologies such as cancer, type 2 diabetes, and adult cardiovascular diseases [39]. TGF-beta is a well-established master regulator of fibrosis [40], which also plays a role in smooth muscle differentiation and proliferation [41]. TGF-beta mediates an SMC switch from the synthetic to contractile phenotype [42]. In the vasculature, elevated TGF-beta/Smad3 promotes SMC proliferation and intimal hyperplasia following endovascular injury [43], and crosstalk has been established between TGF-beta and Wnt/beta catenin pathways, promoting SMC proliferation [44]. Our data indicate that similar crosstalk might exist in LUTD of different origins, leading to bladder hypertrophy. Current findings underline the involvement of Wnt, TGF-beta, and FGF activity for pathologic bladder remodeling, paving the way to novel pharmacological interventions. 

## 4. Materials and Methods

### 4.1. Animals

Female C57Bl/6J mice (*n* = 60) were obtained from Charles River Laboratories (France). They were housed at a 12/12 light–dark cycle, and food and water were available ad libitum, first in conventional cages until the age of 11 weeks, or single-housed after implanting catheter and EMG electrodes to avoid implant damage by cage-mate biting. Cages with inserted dividers (green line, Tecniplast, Buguggiate, Italy) were used, allowing interaction by sound and smell and reducing social deprivation, complying with the 3R principles of animal experimentation. 

### 4.2. Study Approval

The animal experiments were performed in accordance with the relevant Swiss laws and approved by the Veterinary Commission for Animal Research of the Canton of Berne, Switzerland (License Nr BE 53/18).

### 4.3. Catheter and Electrode Implantation 

Mice in the pBOO study were implanted with a catheter, and mice in the SCI study were implanted with a catheter and EUS-EMG electrodes. The implantation surgery was performed as previously described [20]. Briefly, mice were anesthetized with a mixture of medetomidine (0.5 mg/kg, Domitor, Orion Corporation, Espoo, Finland), midazolam (5 mg/kg, Dormicum^®^ Midazolamum 5 mg/mL, Roche Pharma (Schweiz) AG, Reinach, Switzerland), and fentanyl (50 µg/kg, Fentanyl Sintetica, Sintetica S.A., Mendrisio, Switzerland) injected subcutaneously, and midline laparotomy performed to expose the bladder. A catheter with a flared end (Intravascular PE-10 tubing, SAI Infusion Technologies, Lake Villa, Illinois, USA) was implanted into the bladder dome with a purse string suture (6-0 PROLENE^®^, 8807H, Ethicon, Somerville, NJ, USA) and tunneled subcutaneously to the neck of the animal, externalized, and fixed to an infusion harness (SMH, SAI Infusion Technologies, Lake Villa, IL, USA). The animal wore the harness throughout the experiment. In the SCI study, mice additionally were implanted with two electrodes affixed on either side of the EUS to the lateral fat tissue and one ground electrode to the abdominal muscle. The electrodes were tunneled subcutaneously into the animal’s neck, exteriorized, soldered to a connector (850-10-050-10-001101, Preci-Dip, Delémont, Switzerland), and affixed to the harness. Daily assessments of bodyweight, physical appearance (mobility, bite wounds), and signs of pain served to monitor the animal’s health.

### 4.4. Partial Bladder Outlet Obstruction (pBOO) Surgery

pBOO or sham surgery was performed in mice one week after catheter implantation and after baseline UDI. Mice were anesthetized by subcutaneous injection of medetomidine, midazolam, and fentanyl, as described above. A lower midline abdominal incision was made, and the urethra was exposed. A catheter (Intravascular PE-10 tubing, SAI Infusion Technologies, Lake Villa, IL, USA) was inserted into the urethra, and a ligature (6-0 Permahand^®^ silk suture, 639H, Ethicon, Somerville, NJ, USA) was tied around the urethra to partially obstruct the bladder outlet, followed by removal of the catheter. Daily assessments of bodyweight, physical appearance (mobility, bite wounds), and signs of pain served to monitor the animal’s health. 

### 4.5. Spinal Cord Injury (SCI) Surgery 

Mice received SCI or sham surgery one week after catheter and electrode implantation and after baseline UDI. Mice were anesthetized by subcutaneous injection of medetomidine, midazolam, and fentanyl, as described above. The harness was removed to expose the upper back of the animal, and a laminectomy was performed at thoracic level 8. The spinal cord was transected completely with microsurgical scissors. In mice belonging to the sham group, no laminectomy and transection of the spinal cord were performed. Mice with an SCI were unable to void spontaneously, so their bladders were manually expressed twice a day throughout the rest of the experiment. Daily assessments of bodyweight, physical appearance (mobility, bite wounds), and signs of pain served to monitor the animal’s health. 

### 4.6. Urodynamic Investigation 

UDI recordings were performed as previously described [20]. Briefly, mice were placed into an adapted restrainer (pBOO study: HLD-MS-T, Kent Scientific Corporation, Torrington, CT, USA; SCI study: KN-326-4, Natsume Seisakusho Co., Ltd., Tokyo, Japan), which allowed voids to pass onto the scale placed underneath. The scale recorded voided volumes with a frequency of 5 Hz. The catheter was connected to an infusion pump (R-100EC, Razel Scientific Instruments, Saint Albans, VT, USA) with an in-line pressure transducer (TDR-100-1, Med Associates Inc., Fairfax, VT, USA). The pressure signal was amplified (CSG-6080, Catamount R & D Inc., St. Albans, VT, USA) and sampled at 500 Hz and 5000 Hz in the pBOO and SCI study, respectively. In the SCI study, the electrodes were connected to the amplifier (Model 1700, Science Products GmbH, Hofheim, Germany), and the EMG signal was recorded at 5000 Hz, band-pass filtered between 10 and 10,000 Hz, and the gain was set to ×1000. The power line interference was reduced by including a notch filter. All signals were sampled through a data acquisition board (NI USB-6211, National Instruments, Austin, TX, USA) and recorded on the computer with a self-programmed LabVIEW v2012 (National Instruments, Austin, TX, USA) application. During UDI, the bladder was infused with physiological saline (NaCl 0.9%) at a rate of 10 µL/min in the pBOO study and 20 µL/min in the SCI study. P_ves_, the EUS-EMG (in the SCI study), and P_void_ were recorded simultaneously. Once stable micturition cycles have been established in the pBOO study and in baseline and sham recordings of the SCI study, the recording was continued for at least 3 micturition cycles. In mice with SCI, the bladders were first manually expressed, and UDI was recorded for at least 25 min. 

### 4.7. UDI Analysis

Urodynamic data analysis was performed using MATLAB^®^ as previously described [20]. Three micturition cycles were analyzed per animal per week. P_max_ was the highest P_ves_ observed in the micturition cycle. P_thresh_ was the P_ves_ when the derivative of the smoothed pressure signal (moving average over 0.1 s, which corresponded to 50 in the pBOO study and 500 in the SCI study) for the first time reached 65% of the maximal derivative within 20 s prior to P_max_. Filling volume (V_filling_) was the volume being infused into the bladder from the beginning of the cycle until P_thresh_ was reached. When no micturition cycles were observed, P_ves_ was normalized to the lowest value of the entire recording, and P_max_ was the highest P_ves_ observed. 

In the SCI study, EUS-EMG was recorded simultaneously with P_ves_. The EUS activity during voiding was analyzed as previously established [20]. Briefly, the Hilbert-Huang transform was implemented to analyze the time-frequency components of the EUS-EMG signal. The time period from P_thresh_ to P_max_ was divided into 5 equal parts, and the middle part was considered “during voiding”. The total energy at frequencies 0–500 Hz was calculated per section (before, during, and after voiding) and normalized to the duration of the respective section. The EMG activity is shown as a percentage between these three sections. The percentage of voiding contractions was calculated, which were defined as contractions during which the EUS-EMG showed a “v-shape”. V-shape activity means that the EUS showed the lowest activity in the middle section. Animals were excluded from this analysis when less than 5 contractions could be analyzed in those 10 min. 

### 4.8. Euthanasia and Bladder Harvest

After the last UDI, mice were deeply anesthetized with an intraperitoneal injection of overdosed Esconarkon^®^ ad. us. vet. (150 mg/kg) diluted 1:3 in NaCl 0.9%. As soon as the respiratory reflex was arrested and the pedal reflex was absent, mice were transcardially perfused (7.6 mL/min) with chilled (4 °C) NaCl 0.9 for two min. The bladder was harvested, and electrode placement was controlled in the mice of the SCI study. The bladder was weighted, and pieces were prepared for further processing: Half of the bladder was fixed in 4% paraformaldehyde (Sigma-Aldrich Chemie, Buchs, Switzerland) overnight at 4 °C and transferred to PBS with 30% sucrose (84097-250 G, Sigma-Aldrich Chemie, Buchs, Switzerland). Bladders were stored at 4 °C until processing for histological examination. Half of the bladder in the pBOO study and a quarter of the bladder in the SCI study were immersed in RNAlater (R0901, Sigma-Aldrich) and stored until RNA isolation and transcriptome analysis. Bladder-to-bodyweight ratio was calculated by dividing the bladder weight by the mean of the animal’s bodyweight measured on the last three consecutive days.

### 4.9. Bladder Histology

Fixed bladders stored in PBS with 30% sucrose were enclosed in TissueTek^®^ O.C.T.^TM^ compound (Sakura, Staufen, Germany) and cryosectioned at 5 µm thickness. Sections were stained with Masson’s trichrome stain (ab150686, Abcam), and images were taken with a slide scanner (Pannoramic 250 Flash II, 3DHistech, Budapest, Hungary). Using Image J [45], one section per mouse was analyzed to assess bladder fibrosis. The ratio of collagen tissue to the rest of the bladder tissue was calculated within the detrusor muscle and in a homogeneous section through all bladder layers. Two experimenters performed the analysis in a blinded fashion, and the mean values of these analyses were used for statistical analysis. 

### 4.10. Bladder Transcriptome Analysis

Total RNA was isolated from bladder portions previously immersed in RNAlater and stored at −80 °C using mirVana^TM^ RNA Isolation Kit as described previously [46]. cDNA was produced using the High Capacity cDNA Reverse Transcription Kit (4368814, Applied Biosystems), and the content of selected mRNAs was assessed by RT-qPCR. The following panel of inflammatory TagMan^TM^ Gene Expression Assays (Applied Biosystems) was used: Ccl2 (Mm00441242_m1), Nlrp3 (Mm00840904_m1), Adgre1 (Mm00802529_m1), Tnfaip3 (Mm00437121_m1), Ctgf (Mm01192933_g1), CD68 (Mm03047343_m1), Irf1 (Mm01288580_m1), Ccr2 (Mm00438270_m1), Hif1a (Mm00468869_m1), Ldha (Mm01612132_g1), and Vegfa (Mm00437304_m1). Tested mRNA Ct values were normalized to Ct values of 18s, and log2 fold changes were calculated. 

A total of 2 µg of bladder RNA was used for transcriptome analysis by NGS, as described previously. Reads were aligned and normalized, and counting per gene was performed using HTSeq v.0.5.4p3. The Bioconductor packages DESeq2 and edgeR (Bioconductor version: Release (3.2)) were used to identify DEGs. A threshold of 0.15 was chosen for the FDR-adjusted *p*-value. For PCA on log2 fold change of mRNA expression, the ‘prcomp’ function implemented in R (R Core Team, 2016), rgl, and scatterplot3d R package were used. Differentially regulated genes with *p*-value < 0.1 or adjusted *p*-value < 0.1 were included in the pBOO and SCI studies, respectively. In order to achieve better numerical accuracy, the PCA was performed by a singular value decomposition of the (centered and scaled) data matrix. To gain biological insight into the DEGs, Gene Ontology (GO) Over-Representation Analysis (ORA) [47] methods were used. ClusterProfiler (version 3.18.1) package [48] in R was used to perform GO-ORA on biological processes and cellular compartment terms associated with DEGs. Results obtained at a threshold of *p*-value < 0.1 were considered statistically significant. Gene enrichment data from the pathway analysis dataset was condensed and visualized by creating a word cloud using Wordle.net and the Word cloud R package. The incidence of a gene in the pathway analysis dataset is represented by its font size. 

### 4.11. Statistical Analysis

Body weight, bladder-to-bodyweight ratio, bladder collagen content, and qPCR data were statistically analyzed using an unpaired t-test comparing each disease group to its corresponding sham group. Three micturition cycles were analyzed per mouse, and mean values were calculated. Urodynamic parameters within each group and among groups at the same time point were compared using a one-way repeated-measures ANOVA followed by Bonferroni’s post hoc testing. Statistical analyses were performed using the Stata software (version 14.2, College Station, StataCorp, TX, USA) and GraphPad Prism (version 7.04, GraphPad Software, Inc., San Diego, CA, USA). Differences were considered significant when *p* < 0.05. 

## 5. Conclusions

In summary, our study revealed the dire consequences of chronic NLUTD and DSD on bladder morphology and function in a mouse model of SCI with complete spinal cord transection at the level of T8. SCI induced profound changes in neuronal signaling and muscle contractility and caused significant bladder fibrosis 8 weeks after injury. These changes were robust and highly different from pBOO-induced bladder remodeling of similar duration, which showed signs of functional compensation on the background of bladder hypertrophy and inflammation. Both types of LUTD share the processes of growth, proliferation, fibrosis, and the activation of Wnt and TGF-beta signaling. Current findings support our earlier observations concerning the importance of Wnt signaling for the regulation of smooth muscle contractility. These results open up the possibility of developing novel pharmacological tools to prevent precipitous bladder fibrosis and maintain organ function during BOO. 

## Figures and Tables

**Figure 1 ijms-24-02451-f001:**
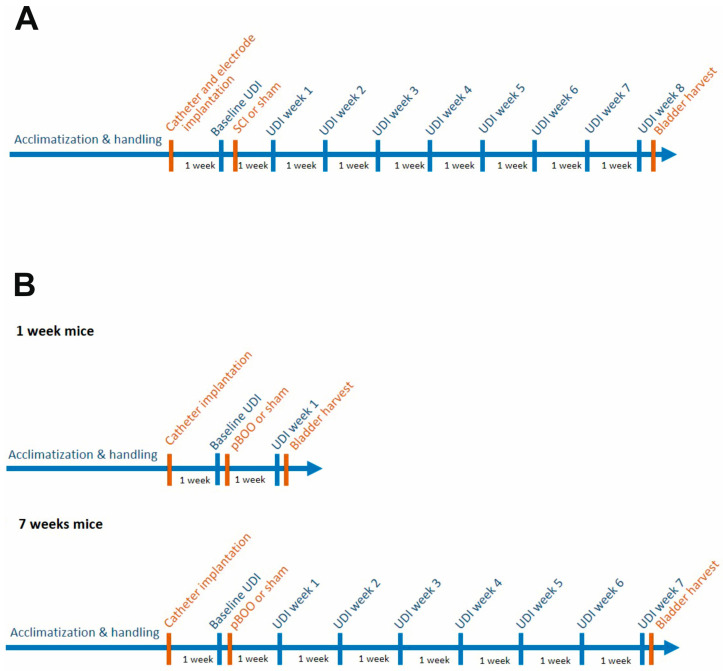
Study design. (**A**) SCI study, (**B**) pBOO study.

**Figure 2 ijms-24-02451-f002:**
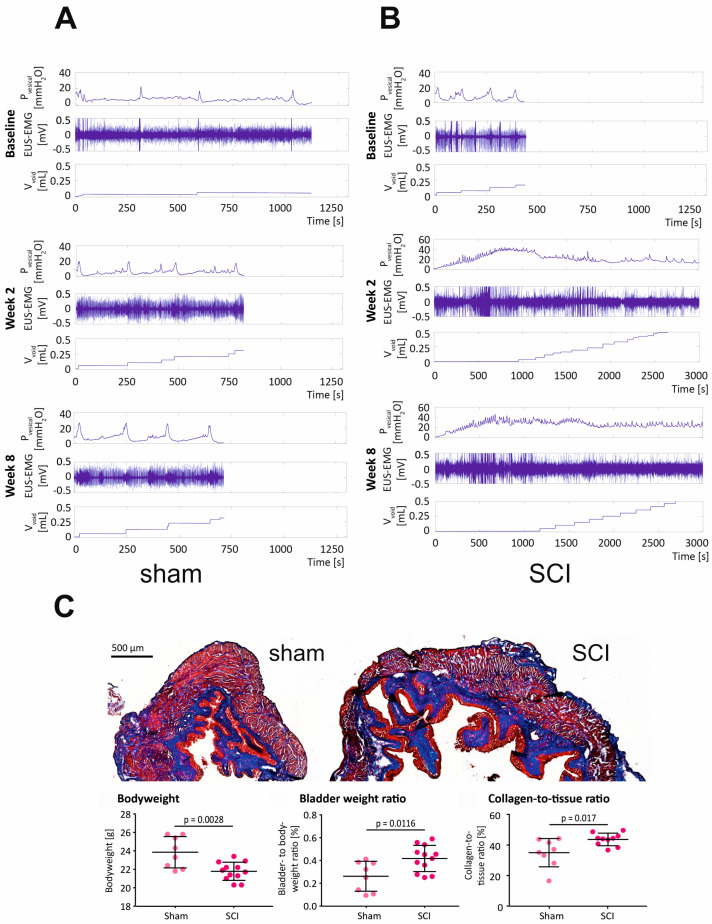
Functional and morphological alterations in the bladders of SCI mice. Representative UDI and EUS-EMG recordings in sham (**A**) and SCI (**B**) mice before injury (baseline) and at 2 and 8 weeks. (**C**) Bladder morphology was evaluated by Masson’s trichrome stain-ing, bladder-to-bodyweight ratio, and % of collagen in the bladder wall were calculated.

**Figure 3 ijms-24-02451-f003:**
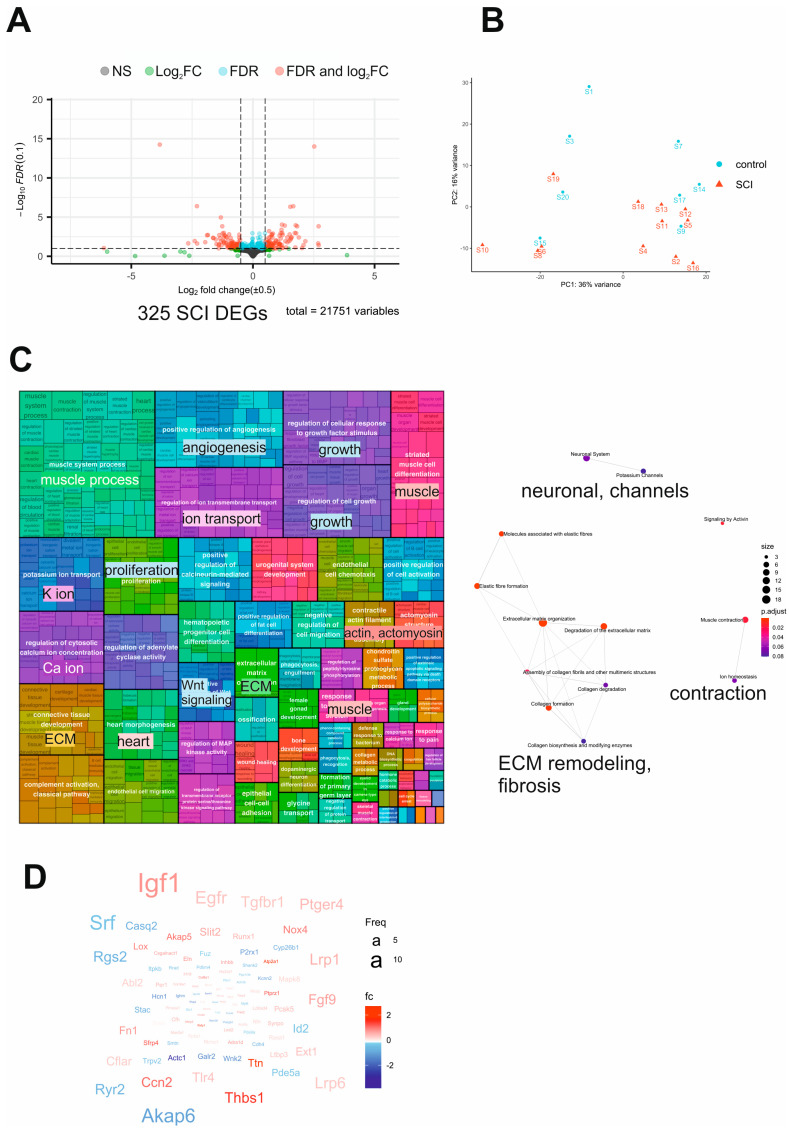
Transcriptome analysis of SCI mouse bladders. (**A**) Volcano plot of all mRNAs. Upregulated mRNAs are visualized in red color and downregulated in green. mRNAs with *p*-value < 0.05, absolute log2 fold change > 0.5 were selected for further analysis. (**B**) PCA of the RNA sequencing data in a 2D graph of PC1 and PC2 based on normalized read counts of all mRNA (325) with adjusted *p*-value < 0.05, absolute log2 fold change > 0.5, and mean of read counts > 50 reads. The bi-plot shows samples as labeled dots. (**C**) A treemap view of GO-term clusters (Biological Processes (BP)) and Reactome pathway analysis. (**D**) Word cloud of upregulated (in red) and downregulated (in blue) mRNAs, font size corresponding to the frequency of appearance in GO-BP.

**Figure 4 ijms-24-02451-f004:**
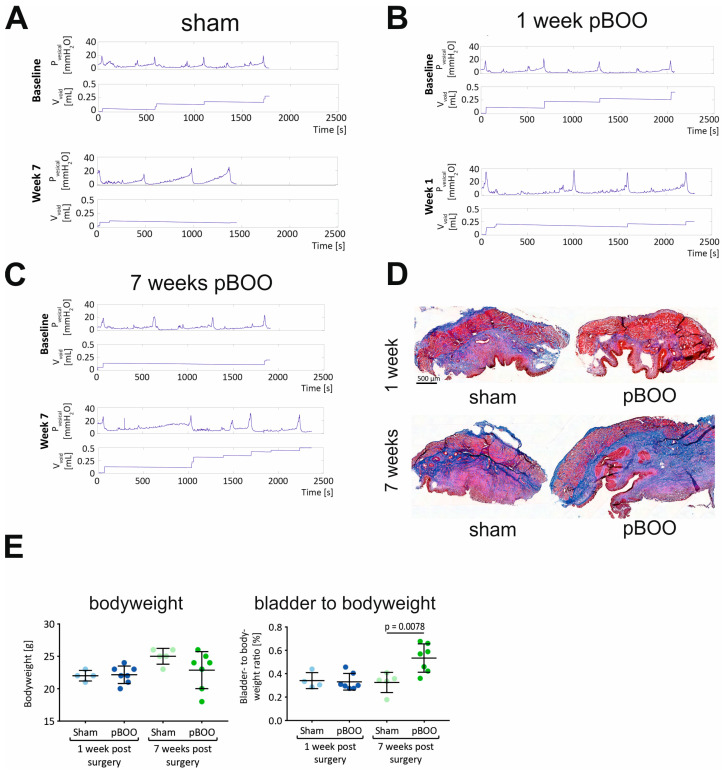
Functional and morphological alterations in the bladders of pBOO mice. Representative UDI recordings in sham (**A**) and pBOO at 1 week (**B**) and pBOO at 7 weeks (**C**) post-obstruction. Bladder morphology was evaluated by Masson’s trichrome staining (**D**), bladder-to-bodyweight ratio was calculated for each group (**E**).

**Figure 5 ijms-24-02451-f005:**
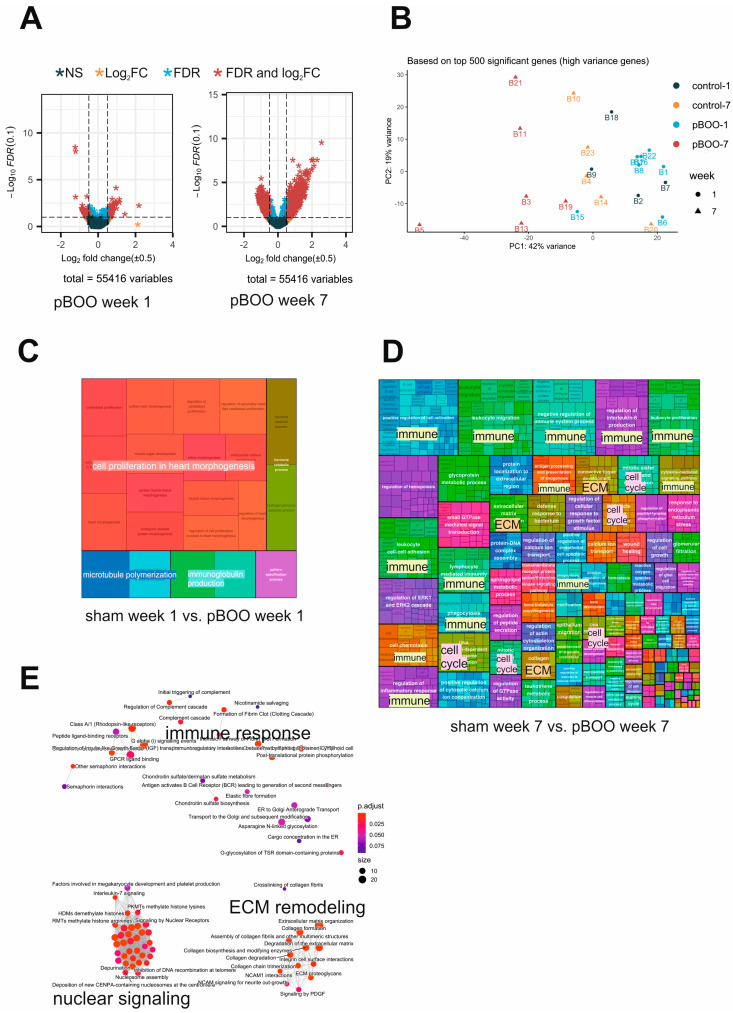
Transcriptome analysis of pBOO mouse bladders. (**A**) Volcano plot of all DEGs at week 1 and week 7 vs. respective shams. mRNAs with *p*-value < 0.05, absolute log2 fold change > 0.5 (in red) were selected for further analysis. (**B**) Principal component analysis of the RNA sequencing data in a 2D graph of PC1 and PC2 based on normalized read counts of all mRNA with adjusted *p*-value < 0.05, absolute log2 fold change > 0.5 and mean of read counts > 50 reads. The bi-plot shows samples as labeled dots. (**C**) A treemap view of GO-term clusters (Biological Processes) at 1 week and 7 weeks (**D**) pBOO, and Reactome pathway analysis (**E**) at 7 weeks pBOO.

**Figure 6 ijms-24-02451-f006:**
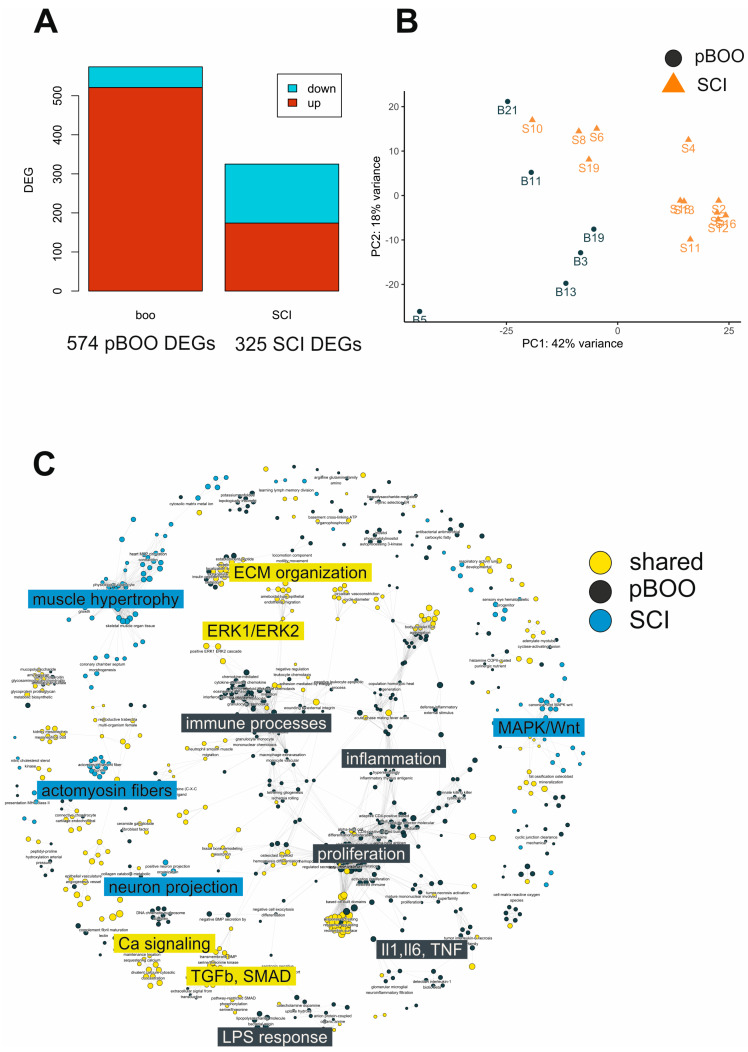
Comparative analysis of pBOO and SCI bladder transcriptomes. (**A**) Number of DEGs in pBOO (7 weeks) and SCI (8 weeks) bladders. (**B**) PCA of the RNA sequencing data of pBOO and SCI bladders. (**C**) Enrichment map of the shared (in yellow), SCI-specific (in blue), and pBOO-specific (in gray) biological processes.

**Figure 7 ijms-24-02451-f007:**
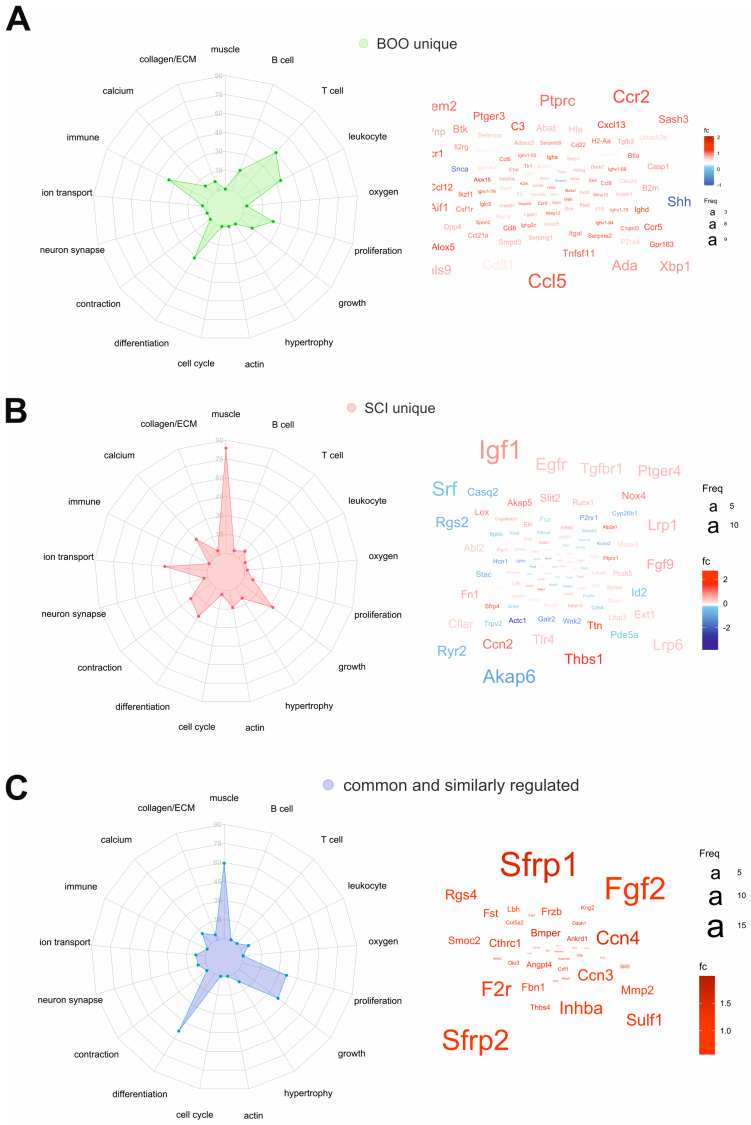
Unique and common biological processes in pBOO and SCI bladders. Radar graphs for major BPs and word clouds for DEGs involved in the majority of BPs are shown. Analysis was performed using (**A**) DEGs only regulated in pBOO bladders (pBOO unique), (**B**) DEGs only regulated in SCI bladders (SCI unique), and (**C**) 46 DEGs, common and similarly regulated in pBOO and SCI transcriptomes.

**Table 1 ijms-24-02451-t001:** P_max_ in SCI vs. sham mice.

P_max_	Baseline	Week 1	Week 2	Week 3	Week 4	Week 5	Week 6	Week 7	Week 8
**sham**	18.82 ± 2.4	20.37 ± 2.35	21.78 ± 7.44	20.21 ± 3.07	21.07 ± 4.09	22.14 ± 4.51	24.11 ± 4.06	25.26 ± 6.77	29.21 ± 12.04
**SCI**	18.43 ± 4.13	41.49 ± 10.77 **	36.18 ± 7.09 **	39.77 ± 7.85 ***	35.82 ± 8.02 **	35.14 ± 7.47 **	40.68 ± 9.99 **	41.29 ± 8.6 **	39.13 ± 6.71 **

** week 1: *p* = 0.004; ** week 2: *p* = 0.004; *** week 3: *p* < 0.001; ** week 4: *p* = 0.005; ** week 5: *p* = 0.009; ** week 6: *p* = 0.009, ** week 7: *p* = 0.009. P_max_ is the highest intravesical pressure observed in the micturition cycle.

**Table 2 ijms-24-02451-t002:** P_max_ in pBOO vs. sham mice.

P_max_	Baseline	Week 1	Week 2	Week 3	Week 4	Week 5	Week 6	Week 7
**Sham 1 wk**	18.7 ± 3.14	19.86 ± 2.6						
**pBOO 1 wk**	16.9 ± 2.7	29.4 ± 4.6 **						
**Sham 7 wks**	17.9 ± 5	21.09 ± 3	25.2 ± 5.3	23.6 ± 5.3	26.26 ± 7.5	22.3 ± 8.5	21.7 ± 6.6	22.12 ± 2.5
**pBOO 7 wks**	16.02 ± 4.34	37.5 ±12.7 *	30.4 ± 6.7	29 ± 14.9	21 ± 7.2	21.45 ± 4.5	21.43 ± 5.1	20.5 ± 7.8

** pBOO 1 wk: *p* = 0.006; * pBOO 7 wks: *p* = 0.032. P_max_ is the highest intravesical pressure observed in the micturition cycle.

## Data Availability

The lists of differentially expressed genes used for downstream analysis and regulated genes involved in signaling pathways are available as Appendix A. The mRNA-seq datasets are deposited in the European Nucleotide Archive (ENA) under project accession number PRJEB58912.

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
