# Peer review of "Molecular Characterization of Non-Neurogenic and Neurogenic Lower Urinary Tract Dysfunction (LUTD) in SCI-Induced and Partial Bladder Outlet Obstruction Mouse Models"

_ijms, 2023, doi:10.3390/ijms24032451_

Round 1
Reviewer 1 Report
In the study by von Siebenthal and collaborators, the authors collected bladder tissue from animals with lower urinary tract dysfunction (LUTD) of neurogenic and non-neurogenic origin. Mice were submitted to spinal cord transection as a model of spinal cord injury (SCI) or partial bladder outlet obstruction (pBOO) and submitted to weekly urodynamic evaluation during the experimental periods, after which they were euthanized, and bladders collected. The tissue was processed for transcriptomic analysis and maps of differentially expressed genes (DEGs) were obtained. The authors found a collection of genes similarly altered in both types of LUTD, mostly related to remodelling of the bladder wall. They also found specific genetic signatures for neurogenic and non-neurogenic LUTD, with changes in SCI bladders pointing to muscle hypertrophy and changes in nerve fibres while changes in pBOO indicate the activation of inflammatory processes and immune response. Therefore, although urodynamic records in both conditions may present similarities, the underlying molecular mechanisms are not necessarily the same, although some pathways may be simultaneously relevant in neurogenic and non-neurogenic LUTD.
This is a relevant and well conducted study, that deserves to be published after some points have been improved:
- My main concern resides in the SCI group. N-LUTD was only investigated at a late time-point of disease progression, while it is know acknowledged that changes are initiated much earlier, likely immediately after spinal injury. There is no SCI group for an earlier time-point than 7 weeks. In fact, the fact that bladder changes are also set in motion very early after after pBOO has been recognized by the authors as 1- and 7-weeks post-BOO were examined.
- The study would be enriched if the RT-cPCR experiments, performed to validate the NGS analysis, would be included in the main figures. However, this validation was only performed for the pBOO group (late stage). Why not 1-week pBOO? And how about the SCI group? Were results validated?
- A minor point has to do with the size of lettering in figures. Is it possible to increase the size? For example, in figure 5A I am making an educated guess about what reads on top of the graph. This happens in many figures, including supplementary ones.
- Another minor point resides on presentation of intravesical pressure. Why present it as a graph in table 1 and a graph in figure 4?
Author Response
We are grateful to this Reviewer for the insightful comments, which have helped us to improve the quality of our manuscript.
The point-by-point replies to the comments are listed below:
- My main concern resides in the SCI group. N-LUTD was only investigated at a late time-point of disease progression, while it is know acknowledged that changes are initiated much earlier, likely immediately after spinal injury. There is no SCI group for an earlier time-point than 7 weeks. In fact, the fact that bladder changes are also set in motion very early after after pBOO has been recognized by the authors as 1- and 7-weeks post-BOO were examined.
We wholeheartedly agree with the Reviewer that the early time points after SCI are very important and yield a wealth of information about the potential drivers of bladder remodelling. In fact, we have recently completed such a study in SCI rats, and the data are very interesting. We did not include the 1 week time point in the mouse study, because the catheter and EMG electrode implantation in mice proved to be quite challenging, and we did not want to introduce an additional group and risk not having enough animals towards the end of the study. We were also concerned about the mice still being in the spinal shock phase after 1 week of SCI, and thus not yet developing the “obstructed” phenotype, comparable to 1 week of pBOO. Despite this, we are convinced that the strength of our work lies in the comparative analysis of the bladder transcriptomes in NLUTD vs. BLUTD, as we attempted to do at 7/8 weeks after introducing obstruction.
- The study would be enriched if the RT-cPCR experiments, performed to validate the NGS analysis, would be included in the main figures. However, this validation was only performed for the pBOO group (late stage). Why not 1-week pBOO? And how about the SCI group? Were results validated?
The current consensus is that QPCR validation of the NGS results is not necessary, unless done in a bigger cohort of unsequenced samples. As our sample size was not large, we sequenced them all. Ironically, QPCR of the pBOO samples presented in Fig. S2 was carried out before NGS, because we were bewildered by the functional compensation after 7 weeks of obstruction. Results of QPCR, together with the obvious changes of bladder-to-bodyweight ratio convinced us that the pBOO bladders underwent remodelling and we proceeded with RNAseq, which proved quite informative.
In contrast, the functional changes in the SCI bladders were very apparent at UDI; hence we preformed NGS immediately.
- A minor point has to do with the size of lettering in figures. Is it possible to increase the size? For example, in figure 5A I am making an educated guess about what reads on top of the graph. This happens in many figures, including supplementary ones.
Done. Figures 3, 4, 5 and 3S have been changed
- Another minor point resides on presentation of intravesical pressure. Why present it as a graph in table 1 and a graph in figure 4?
Pmax results are now presented as Tables: Table 1 for SCI, Table 2 for pBOO, upon request of this and the other Reviewer.

Reviewer 2 Report
Authors firstly compared the transcriptome characteristics of bladder between SCI and BOO mice model, which provided intriguing information for urological research. However, rigid amendments should be achieved before reconsideration.
1. Please explain why the observation timescale in SCI mice and pBOO mice are different, and then explain whether the procedure that authors declared in abstract is correct (UDI assessments were performed weekly during 8 weeks).
2. Please explain how the numbers of mice in each subgroup were determined, e.g., 8 mice in sham and 19 in SCI.
3. Please explain how the infusion rates were set in 10 µl/min in the pBOO and 20 µl/min in the SCI.
4. What is the meaning of orange color in figure 2 A and B? Does that represent the second micturition cycle?
5. The observation timescales of UDI were different in each week and each subgroup, each figure only showed the three cycles, which were not convincing.
6. Line 514, 515. “In mice with SCI, the bladders were first manually expressed…” Where are the time points that authors express bladder? In the situation of figure 2 B, the micturition cycle is hard to be distinguished. The frequent leaks of urine should happen under the situation of dyssynergia, so that are definitely not three cycle in Fig2B week2 and week8.
7. Figure 4 A need clear labels. There are 5 UDI figures. Which one is sham and which one is pBOO? What do the left labels (1 week and 7 week) represent? There are labels (week1 and baseline) next to the left label (1 week). These are very confusing. Figure 4 B. What do the blue, purple, light green, and green bars represent?
8. The pixel value of images in treemaps could elevated or just enlarge the labels.
9. The influence of Wnt and TGF-β signalings on smooth muscle should be discussed. Their effects on detrusor should be further described and discussed, the related previous evidence should be referred, rather than authors’ previous works only. Authors merely mentioned the up-regulation of pathways in results, which provided no significance to the readership. The same situation also happened to the result of FGF2.
10. Line 50. The factors that cause LUTD are far more than just BOO. Authors may introduce more comprehensive factors in this section before highlighting your aims.
11. Line 71. It should be “HIF1α”.
12. Line 80 to 83. Please describe specifically. Focal adhesion of what. What kind of regulation? What type of immune response?
13. Some confusing descriptions and typo could be amended.
Line 22. “weekly during 8 weeks followed by bladder harvest…” could be “weekly during 8 weeks and were followed by bladder harvest for histological and transcriptome analysis”.
Line 61. Perhaps “progresses to hypertrophy, and afterwards fibrosis that ultimately leads to bladder decompensation”.
Line 88. “at” mRNA level
Line 102. and “was” correlated to gene expression…
Line 113, 127. “bladders were harvested…”
Author Response
We are grateful to this Reviewer for the suggestions and comments.
The point-by-point replies to are listed below:
Authors firstly compared the transcriptome characteristics of bladder between SCI and BOO mice model, which provided intriguing information for urological research. However, rigid amendments should be achieved before reconsideration.
- Please explain why the observation timescale in SCI mice and pBOO mice are different, and then explain whether the procedure that authors declared in abstract is correct (UDI assessments were performed weekly during 8 weeks).
We stopped pBOO after 7 weeks because the Covid-19 pandemic and ensuing lock-downs mandated termination of all animal experiments in our Animal Facility.
- Please explain how the numbers of mice in each subgroup were determined, e.g., 8 mice in sham and 19 in SCI.
Numbers were based on expected dropout rate. We expected less dropouts in the sham group. Based on previous experiments (pBOO study and rat SCI studies) we additionally expected a higher variability in UDI parameters in the intervention group. Originally, we planned 10 mice in the sham group and 20 in SCI group, but lost 3 during implantation surgery.
- Please explain how the infusion rates were set in 10 µl/min in the pBOO and 20 µl/min in the SCI.
We used an infusion pump. For pBOO we used 10 ul/min, which corresponds to the lowest infusion rate reported in the literature. We used 20 ul/min in SCI to reduce observation time and stress for the animals, since an increased bladder capacity was expected.
- What is the meaning of orange color in figure 2 A and B? Does that represent the second micturition cycle?
Yes
- The observation timescales of UDI were different in each week and each subgroup, each figure only showed the three cycles, which were not convincing.
We are showing three representative micturition cycles in this figure, because we analysed 3 micturition cycles per measurement per animal at each time point. The difference in the duration of 3 cycles represents the intra- and inter-individual variability. The figure is not showing entire measurements, e.g. the beginning of the measurement, until stable micturition cycles were established, is not shown. Timescales in Figure 4 are consistent (2500 s). In Figure 2 timescales are consistent for sham animals and baseline of SCI animals (1250 s). Observation time is prolonged for SCI animals after injury to show that micturition cycles were not established (3000 s).
- Line 514, 515. “In mice with SCI, the bladders were first manually expressed…” Where are the time points that authors express bladder? In the situation of figure 2 B, the micturition cycle is hard to be distinguished. The frequent leaks of urine should happen under the situation of dyssynergia, so that are definitely not three cycle in Fig2B week2 and week8.
Bladder expression was done before connecting mice to the UDI station, thus before the recording, to ensure the bladders were empty before the infusion began.
- Figure 4 A need clear labels. There are 5 UDI figures. Which one is sham and which one is pBOO? What do the left labels (1 week and 7 week) represent? There are labels (week1 and baseline) next to the left label (1 week). These are very confusing. Figure 4 B. What do the blue, purple, light green, and green bars represent?
Figure 4 has now been changed, the graph removed and data shown as Table 2.
- The pixel value of images in treemaps could elevated or just enlarge the labels.
Done, new Figures 3, 5, 3S
- The influence of Wnt and TGF-β signalings on smooth muscle should be discussed. Their effects on detrusor should be further described and discussed, the related previous evidence should be referred, rather than authors’ previous works only. Authors merely mentioned the up-regulation of pathways in results, which provided no significance to the readership. The same situation also happened to the result of FGF2.
Extended discussion has now been added (lines 436-462)
- Line 50. The factors that cause LUTD are far more than just BOO. Authors may introduce more comprehensive factors in this section before highlighting your aims.
Introduction has been re-written to accommodate this point (lines 51-55)
- Line 71. It should be “HIF1α”.
Corrected
- Line 80 to 83. Please describe specifically. Focal adhesion of what. What kind of regulation? What type of immune response?
- Some confusing descriptions and typo could be amended.
Line 22. “weekly during 8 weeks followed by bladder harvest…” could be “weekly during 8 weeks and were followed by bladder harvest for histological and transcriptome analysis”.
Line 61. Perhaps “progresses to hypertrophy, and afterwards fibrosis that ultimately leads to bladder decompensation”.
Line 88. “at” mRNA level
Line 102. and “was” correlated to gene expression…
Line 113, 127. “bladders were harvested…”
Corrected / explained as requested

Round 2
Reviewer 2 Report
The overall issues were well-amended by authors. This research provided significance to further reveal the mechanism of bladder remodeling in pBOO and SCI mice model. Some points could be further amended.
1. Reviewer suggests the orange color in figure 2 B (SCI group, week 2, week 8) can be changed just to blue, since it is unclear to discriminate the second micturition cycle. The data of the V of void beneath EUS-EMG showed several times of urine outflows, which suggested there were more than three cycles. Authors may just add further descriptions about this situation.
2. In figure 4. The data of V void in fig 4 A (week 7), B (week 1), C (baseline) were unchanged. Please explain this situation or just replace those with data from the other mice.
3. Please check the value in tables. In table 2, sham group, week 7, it is 2.12 ± 2.5.
4. Define Pmax in table legend again.
5. Line 54. Revise “Multiple Sclerosis”, just multiple sclerosis.
6. Some misleading descriptions could be revised, which are as follows: Line 33. highlighted FGF2 as a major “up-regulated” transcription factor
Author Response
- Reviewer suggests the orange color in figure 2 B (SCI group, week 2, week 8) can be changed just to blue, since it is unclear to discriminate the second micturition cycle. The data of the V of void beneath EUS-EMG showed several times of urine outflows, which suggested there were more than three cycles. Authors may just add further descriptions about this situation.
Colour of the UDI traces in figures 2 and 4 has been changed back to all blue. Explained and methodology of evaluating EMG in the absence of reliable micturition timing has been added to the text (quoted from our earlier publication) (lines 133-136; 155-159)
- In figure 4. The data of V void in fig 4 A (week 7), B (week 1), C (baseline) were unchanged. Please explain this situation or just replace those with data from the other mice.
Trapping of voids leading to unreliable weight recording explained (lines 227-229)
- Please check the value in tables. In table 2, sham group, week 7, it is 2.12 ± 2.5.
Sorry, that was a typo. Corrected to 22.12
- Define Pmax in table legend again.
Done
- Line 54. Revise “Multiple Sclerosis”, just multiple sclerosis.
done
- Some misleading descriptions could be revised, which are as follows: Line 33. highlighted FGF2 as a major “up-regulated” transcription factor
done
